# Occurrence and Genetic Diversity of Protist Parasites in Captive Non-Human Primates, Zookeepers, and Free-Living Sympatric Rats in the Córdoba Zoo Conservation Centre, Southern Spain

**DOI:** 10.3390/ani11030700

**Published:** 2021-03-05

**Authors:** Pamela C. Köster, Alejandro Dashti, Begoña Bailo, Aly S. Muadica, Jenny G. Maloney, Mónica Santín, Carmen Chicharro, Silvia Migueláñez, Francisco J. Nieto, David Cano-Terriza, Ignacio García-Bocanegra, Rafael Guerra, Francisco Ponce-Gordo, Rafael Calero-Bernal, David González-Barrio, David Carmena

**Affiliations:** 1Parasitology Reference and Research Laboratory, Spanish National Centre for Microbiology, 28220 Madrid, Spain; pamelakster@yahoo.com (P.C.K.); dashti.alejandro@gmail.com (A.D.); BEGOBB@isciii.es (B.B.); muadica@gmail.com (A.S.M.); cchichar@isciii.es (C.C.); miguelanez170777@hotmail.com (S.M.); fjnieto@isciii.es (F.J.N.); 2Departamento de Ciências e Tecnologia, Universidade Licungo, Quelimane 106, Zambézia, Mozambique; 3Environmental Microbial and Food Safety Laboratory, Agricultural Research Service, United States Department of Agriculture, Beltsville, MD 20705-2350, USA; jenny.maloney@usda.gov (J.G.M.); monica.santin-duran@usda.gov (M.S.); 4Animal Health and Zoonosis Research Group (GISAZ), Department of Animal Health, University of Córdoba, 14071 Córdoba, Spain; davidcanovet@gmail.com (D.C.-T.); v62garbo@uco.es (I.G.-B.); 5Veterinary Services, Córdoba Zoo Conservation Centre, 14071 Córdoba, Spain; veterinario.zoocordoba@gmail.com; 6Department of Microbiology and Parasitology, Faculty of Pharmacy, Complutense University of Madrid, 28040 Madrid, Spain; pponce@ucm.es; 7SALUVET, Department of Animal Health, Faculty of Veterinary, Complutense University of Madrid, 28040 Madrid, Spain; r.calero@ucm.es

**Keywords:** *Cryptosporidium*, *Giardia*, *Blastocystis*, *Enterocytozoon bieneusi*, *Balantioides coli*, *Troglodytella*, non-human primates, rats, zoological garden

## Abstract

**Simple Summary:**

Little information is currently available on the epidemiology of parasitic and commensal protist species in captive non-human primates (NHP) and their zoonotic potential. This study investigates the occurrence, molecular diversity, and potential transmission dynamics of parasitic and commensal protist species in a zoological garden in southern Spain. The prevalence and genotypes of the main enteric protist species were investigated in faecal samples from NHP, zookeepers and free-living rats by molecular (PCR and sequencing) methods. A high prevalence of the diarrhoea-causing protists *Giardia duodenalis* and *Blastocystis* sp. (but not *Cryptosporidium* spp.) was observed in captive NHP at the Córdoba Zoo Conservation Centre. NHP can harbour zoonotic genotypes of *G. duodenalis*, *Blastocystis* sp., and *Enterocytozoon bieneusi*. Indeed, strong evidence of the occurrence of *Blastocystis* zoonotic transmission between NHP and their handlers was provided, despite the use of personal protective equipment and the implementation of strict health and safety protocols. Free-living sympatric rats are infected by host-specific species/genotypes of the investigated protists and seem to play a limited role as source of infections to NHP or humans in this setting. The extent of these findings should be confirmed in similar epidemiological surveys targeting other captive NHP populations.

**Abstract:**

Little information is currently available on the epidemiology of parasitic and commensal protist species in captive non-human primates (NHP) and their zoonotic potential. This study investigates the occurrence, molecular diversity, and potential transmission dynamics of parasitic and commensal protist species in a zoological garden in southern Spain. The prevalence and genotypes of the main enteric protist species were investigated in faecal samples from NHP (*n* = 51), zookeepers (*n* = 19) and free-living rats (*n* = 64) by molecular (PCR and sequencing) methods between 2018 and 2019. The presence of *Leishmania* spp. was also investigated in tissues from sympatric rats using PCR. *Blastocystis* sp. (45.1%), *Entamoeba dispar* (27.5%), *Giardia duodenalis* (21.6%), *Balantioides coli* (3.9%), and *Enterocytozoon bieneusi* (2.0%) (but not *Troglodytella* spp.) were detected in NHP. *Giardia duodenalis* (10.5%) and *Blastocystis* sp. (10.5%) were identified in zookeepers, while *Cryptosporidium* spp. (45.3%), *G. duodenalis* (14.1%), and *Blastocystis* sp. (6.25%) (but not *Leishmania* spp.) were detected in rats. *Blastocystis* ST1, ST3, and ST8 and *G. duodenalis* sub-assemblage AII were identified in NHP, and *Blastocystis* ST1 in zookeepers. *Giardia duodenalis* isolates failed to be genotyped in human samples. In rats, four *Cryptosporidium* (*C. muris*, *C. ratti*, and rat genotypes IV and V), one *G. duodenalis* (assemblage G), and three *Blastocystis* (ST4) genetic variants were detected. Our results indicate high exposure of NHP to zoonotic protist species. Zoonotic transmission of *Blastocysts* ST1 was highly suspected between captive NHP and zookeepers.

## 1. Introduction

*Cryptosporidium* spp., *Giardia duodenalis*, and *Entamoeba histolytica* are the most frequently identified protozoan parasites causing diarrhoeal disease in humans globally [1]. Clinical manifestations by these infections vary from self-limiting acute diarrhoea in immunocompetent individuals, to fatal chronic diarrhoea in immunocompromised patients [2]. In addition to these well-known enteric pathogens, other potential diarrhoea-causing protist species, including the Stramenopile *Blastocystis* sp. and the Microsporidia *Enterocytozoon bieneusi*, have gained wide clinical and scientific interest in recent years [3,4]. These parasites are transmitted via the faecal-oral route either directly (i.e., person-to-person) or indirectly (i.e., waterborne or foodborne). Remarkably, most of the species/genotypes of the above-mentioned protists can be zoonotically transmitted [5,6,7,8]. For this reason, assessing the occurrence and genetic diversity of enteric protists in domestic, captive, and free-living animal hosts is essential to ascertaining their transmission dynamics, including the occurrence and directionality of zoonotic events.

*Cryptosporidium* spp., *G. duodenalis*, *Blastocystis* sp., and *E. bieneusi* exhibit extensive intra-species genetic diversity leading to the identification of several genotypes/subtypes with marked differences in host and geographical range. At least 40 valid *Cryptosporidium* species and a similar number of genotypes of unknown species status are currently recognized, with *C. hominis* and *C. parvum* causing most of the infections documented in humans and non-human primate (NHP) species [6,9]. *Giardia duodenalis* is currently regarded as a multi-species complex comprising eight (A to H) distinct assemblages, of which assemblages A and B are frequently reported in humans and NHP [5]. At least 28 subtypes (ST) have been proposed within *Blastocystis* sp. with apparent loose host specificity. Of them, ST1–9 and ST12 have been documented in humans and/or NHP, among other vertebrates [8,10,11]. A recent evaluation of ST1–ST26 subtypes concluded that only 22 of those subtypes (ST1–ST17, ST21, ST23–ST26) should be acknowledged as legitimate subtypes [10], with the remaining six pending confirmation in future investigations. Finally, nearly 500 *E. bieneusi* genotypes have been reported and distributed in 11 genetic groups, of which Group 1 (e.g., Type IV, D, and EbpC) and Group 2 (e.g., BEB4, BEB6, I, and J) include most of the potentially zoonotic genotypes [12].

Little is known about the epidemiology of gastrointestinal protist parasites in captive non-human primates (NHP). In Spain, most of the few studies published to date were based on conventional (microscopy) methods and conducted mainly at the zoological gardens of Almuñecar (Granada) and Barcelona [13,14,15,16,17]. Only a single study attempted to characterize the genetic diversity of *G. duodenalis* in NHP at the Madrid and Valencia zoological gardens [18]. Besides *Cryptosporidium* spp., *G. duodenalis*, *Blastocystis* sp., and *E. bieneusi*, ciliated protists in NHP have been even poorly studied. This is the case of *Balantioides coli*, a zoonotic parasite that primarily infect domestic and wild swine, but has also been reported in NHP including gorillas, chimpanzees, bonobos, hamadryas baboons, and Rhesus macaques [19]. Moreover, the commensal ciliate *Troglodytella abrassarti* has been demonstrated a common finding in the faeces of captive and free-living great apes including eastern and western gorillas, chimpanzees, bonobos, and orangutans [20], but little information is available on its occurrence in captive and free-living lesser apes and monkeys.

This molecular-based epidemiological study aims primarily at assessing the frequency and genetic diversity of generalist and host-adapted enteric protist species in captive NHP and their caretakers at the Córdoba Zoo Conservation Centre (CZCC) in southern Spain, with a special focus on the investigation of potential zoonotic transmission events and their directionality. Secondarily, the same survey was conducted in a free-living sympatric rat population in the same enclosure in order to (i) investigate the role of rodents as transmitters of protist infections to NHP and humans and (ii) assess the suitability of rats as natural reservoirs of *Leishmania* spp.

## 2. Materials and Methods

### 2.1. Study Area

The CZCC extends over 4.5 hectares and include 437 specimens belonging to 102 mammalian, reptilian, and avian species. The CZCC has a small but diverse population of NHP species belonging to 10 genera including *Cebuella* (*n* = 2), *Cercocebus* (*n* = 4), *Cercopithecus* (*n* = 3), *Eulemur* (*n* = 2), *Hylobates* (*n* = 3), *Lemur* (*n* = 5), *Macaca* (*n* = 8), *Mandrillus* (*n* = 4), *Saimiri* (*n* = 3), and *Varecia* (*n* = 2). Individuals of the same species are kept in specific enclosures without contact with other NHP, except members of the Lemuridae family (genera *Eulemur*, *Lemur*, and *Varecia*) that share the same enclosure. All NHP are housed in facilities littered with natural materials such as ground bark or earth. The CZCC has strict health and safety protocols in place to ensure that animals, employees, and visitors have a reduced exposure to risk of infection or injury. Employees routinely use appropriate Personal Protective Equipment when in contact with animals or their faecal material.

### 2.2. Sampling

This cross-sectional study included two sampling periods carried out between December 2018 and January 2019, and between November and December 2019. Fresh faecal samples from NHP were directly collected from the ground at the time of routine cleaning and sanitation of enclosures. Information regarding sex, age, and enclosure sharing with other NHP species was recorded at the time of sampling. In parallel, fresh stool samples were also collected from zookeepers and veterinarians in close contact with NHP that volunteered to participate in the study. Human and NHP stool samples were stored at −20 °C without preservatives at the CZCC Veterinary Laboratory until the end of each sampling campaign, when they were shipped to the Spanish National Centre for Microbiology for downstream molecular analyses.

Taking advantage of an ongoing rodent control campaign carried out at the same time as the present study within the CZCC premises undertaken by the local authorities following European guidelines [21], free-living sympatric rats (*Rattus* spp.) were captured using Tomahawk live traps with bait (chicken, dry dog food, fruit, or peanut butter) (Figure 1A). Most captured rats were identified as brown rats (*Rattus norvegicus*), but differential detection with black rats (*Rattus rattus*) was not possible for younger individuals. Traps were placed in the evening and checked for captures the next morning. Rats were anaesthetized with medetomidine (1 mg/kg) and ketamine (50 mg/kg) and then humanely euthanized by an intracardiac injection of sodium pentobarbitone (Dolethal^®^, Vetoquinol Laboratories, Lure, France) at a dose >150 mg/kg (Figure 1B) [22]. Carcases were frozen at −20 °C and shipped to the Spanish National Centre for Microbiology for necropsy (Figure 1C). After thawing and dissection, the small and large intestine was removed, and the intestinal content extracted for further investigation of enteric protists by molecular methods. Additionally, liver, spleen, and ear skin samples were taken to assess the presence of amastigote forms of *Leishmania* spp.

### 2.3. Epidemiological Questionnaire

A standardized questionnaire (Appendix A) and an informed consent was provided as part of the sampling kit to be completed by the CZCC personnel that volunteered to participate in the survey. Questions included: (i) demographic characteristics, e.g., age and sex, (ii) behavioural habits, e.g., hand and fruit/vegetable washing and whether there have been any occurrence of diarrhoea in the participant, their family members, and/or pets, (iii) work-related potential risk factors, e.g., contact with faecal material from NHP and/or other animal species, being a food handler, and (iv) additional questions on other risk factors, e.g., types of drinking water, use of recreational waters in the two weeks prior to sample collection, had any contact with pets and any recent travel abroad.

### 2.4. DNA Extraction and Purification

Genomic DNA was isolated from about 200 mg of each faecal specimen of human, NHP, or rodent origin by using the QIAamp DNA Stool Mini Kit (Qiagen, Hilden, Germany) according to the manufacturer’s instructions, except that samples mixed with InhibitEX buffer were incubated for 10 min at 95 °C. Additionally, genomic DNA from murine tissues (liver, spleen, and ear skin) was isolated using the Speed Tools DNA Extraction Kit (Biotools, Madrid, Spain). To do so, 10–15 mg of each tissue was homogenized in 100 µl of NET-10 buffer (10 mM NaCl, 10 mM EDTA, 10 mM Tris-HCl pH 8) and digested overnight at 56 °C with 100 µL of BT1 buffer (Biotools) and 20 µL Proteinase K (20 mg/mL). After digestion, genomic DNA was extracted according to the manufacturer’s instructions. In all cases, extracted and purified DNA samples were eluted in 200 μL of PCR-grade water and kept at 4 °C until further molecular analysis. A water extraction control was included in each sample batch processed.

### 2.5. Molecular Detection and Characterization of Giardia Duodenalis

Detection of *G. duodenalis* DNA was achieved using a real-time PCR (qPCR) method targeting a 62-bp region of the gene codifying the small subunit ribosomal RNA (*ssu* rRNA) of the parasite [23]. Amplification reactions (25 μL) consisted of 3 μL of template DNA, 0.5 µM of each primer Gd-80F and Gd-127R, 0.4 µM of probe (Appendix A), and 12.5 μL TaqMan^®^ Gene Expression Master Mix (Applied Biosystems, CA, USA). Detection of parasitic DNA was performed on a Corbett Rotor GeneTM 6000 real-time PCR system (QIAGEN) using an amplification protocol consisting of an initial hold step of 2 min at 55 °C and 15 min at 95 °C followed by 45 cycles of 15 s at 95 °C and 1 min at 60 °C. Water (no-template) and genomic DNA (positive) controls were included in each PCR run.

*Giardia duodenalis* isolates that tested positive by qPCR were subsequently assessed by sequence-based multi-locus genotyping of the genes encoding for the glutamate dehydrogenase (*gdh*), β-giardin (*bg*), and triose phosphate (*tpi*) proteins of the parasite. A semi-nested PCR was used to amplify a 432-bp fragment of the *gdh* gene [24]. PCR reaction mixtures (25 μL) included 5 μL of template DNA and 0.5 μM of the primer pairs GDHeF/GDHiR in the primary reaction and GDHiF/GDHiR in the secondary reaction (Appendix A). Both amplification protocols consisted of an initial denaturation step at 95 °C for 3 min, followed by 35 cycles of 95 °C for 30 s, 55 °C for 30 s, and 72 °C for 1 min, with a final extension of 72 °C for 7 min.

A nested PCR was used to amplify a 511 bp-fragment of the *bg* gene [25]. PCR reaction mixtures (25 μL) consisted of 3 μL of template DNA and 0.4 μM of the primers sets G7_F/G759_R in the primary reaction and G99_F/G609_R in the secondary reaction (Appendix A). The primary PCR reaction was carried out with the following amplification conditions: one step of 95 °C for 7 min, followed by 35 cycles of 95 °C for 30 s, 65 °C for 30 s, and 72 °C for 1 min, with a final extension of 72 °C for 7 min. The conditions for the secondary PCR were identical to the primary PCR except that the annealing temperature was 55 °C.

A nested PCR was used to amplify a 530 bp-fragment of the *tpi* gene [26]. PCR reaction mixtures (50 μL) included 2‒2.5 μL of template DNA and 0.2 μM of the primer pairs AL3543/AL3546 in the primary reaction and AL3544/AL3545 in the secondary reaction (Appendix A). Both amplification protocols consisted of an initial denaturation step at 94 °C for 5 min, followed by 35 cycles of 94 °C for 45 s, 50 °C for 45 s, and 72 °C for 1 min, with a final extension of 72 °C for 10 min.

### 2.6. Molecular Detection and Characterization of Cryptosporidium spp.

The presence of *Cryptosporidium* spp. was assessed using a nested-PCR protocol to amplify a 587 bp fragment of the *ssu* rRNA gene of the parasite [27]. Amplification reactions (50 μL) included 3 μL of DNA sample and 0.3 μM of the primer pairs CR-P1/CR-P2 in the primary reaction and CR-P3/CPB-DIAGR in the secondary reaction (Appendix A). Both PCR reactions were carried out as follows: one step of 94 °C for 3 min, followed by 35 cycles of 94 °C for 40 s, 50 °C for 40 s and 72 °C for 1 min, concluding with a final extension of 72 °C for 10 min.

### 2.7. Molecular Detection and Characterization of Blastocystis sp.

Identification of *Blastocystis* sp. was achieved by a direct PCR protocol targeting the *ssu* rRNA gene of the parasite [28]. The assay uses the pan-*Blastocystis*, barcode primer pair RD5/BhRDr to amplify a PCR product of ~600 bp. Amplification reactions (25 μL) included 5 μL of template DNA and 0.5 μM of each primer (Appendix A). Amplification conditions consisted of one step of 95 °C for 3 min, followed by 30 cycles of 1 min each at 94, 59 and 72 °C, with an additional 2 min final extension at 72 °C.

In *Blastocystis*-positive samples from free-living sympatric rats for which Sanger sequencing data were of suboptimal quality, a next-generation amplicon sequencing strategy was used to identify *Blastocystis* sp. subtypes as previously described [29]. In brief, primers ILMN_Blast505_532F and ILMN_Blast998_1017R were used to generate amplicons. These primers amplify a fragment of the *ssu* rRNA gene of ~500 bp and are identical to Blast505_532F/Blast998_1017R [30] with the exception of containing the Illumina overhang adapter sequences on the 5′ end. Amplicons from two rats were used to prepare sequencing libraries, and final libraries were quantified by Qubit fluorometric quantitation (Invitrogen, Carlsbad, CA, USA) prior to normalization. A final pooled library concentration of 8 pM with 20% PhiX control was sequenced using Illumina MiSeq 600 cycle v3 chemistry (Illumina, San Diego, CA, USA). Paired end reads were processed and analyzed with an in-house pipeline that uses the BBTools package v38.82 [31], VSEARCH v2.15.1 [32], and BLAST + 2.10.1. Briefly, read pairs were merged, filtered for quality and length, denoised, and checked for chimeric sequences. Clustering and the assignment of centroid sequences to operational taxonomic units (OTUs) was performed within each sample at a 98% identity threshold. Only those OTUs with a minimum of 100 sequences were retained and then checked for chimeras once more. OTUs were then blasted against *Blastocystis* references from the National Center for Biotechnology Information (NCBI). Hits below an alignment length of 400 bp were removed.

### 2.8. Molecular Detection and Characterization of Enterocytozoon bieneusi

Detection of *E. bieneusi* was conducted by a nested PCR protocol to amplify the internal transcribed spacer (ITS) region as well as portions of the flanking large and small subunit of the ribosomal RNA gene as previously described [33]. The outer EBITS3/EBTIS4 and inner EBITS1/EBITS2.4 primer sets (Appendix A) were used to generate a PCR product of 390 bp, respectively. Cycling conditions for the primary PCR consisted of one step of 94 °C for 3 min, followed by 35 cycles of amplification (denaturation at 94 °C for 30 s, annealing at 57 °C for 30 s, and elongation at 72 °C for 40 s), with a final extension at 72 °C for 10 min. Conditions for the secondary PCR were identical to the primary PCR except only 30 cycles were carried out with an annealing temperature of 55 °C.

### 2.9. Molecular Differential Detection of Entamoeba histolytica and Entamoeba dispar

Detection and differential diagnosis between pathogenic *E. histolytica* and non-pathogenic *E. dispar* were carried out by a qPCR method targeting a 172-bp fragment of the gene codifying the *ssu* rRNA gene of the *E. histolytica*/*E. dispar* complex [34,35]. Amplification reactions (25 μL) consisted of 3 μL template DNA, 12.5 pmol of the primer set Ehd-239F/Ehd-88R, 5 pmol of each TaqMan^®^ probe (Appendix A), and TaqMan^®^ Gene Expression Master Mix (Applied Biosystems). Cycling conditions and data analysis were as described above for the detection of *G. duodenalis*.

### 2.10. Molecular Detection of Balantioides coli

Detection of *B. coli* was attempted by a direct PCR assay to amplify the complete ITS1–5.8s-rRNA–ITS2 region and the last 117 bp (3’ end) of the *ssu*-rRNA sequence of this ciliate using the primer set B5D/B5RC [36]. PCR reactions (25 μL) consisted of 2 μL of template DNA and 0.4 μM of each primer (Appendix A). PCR conditions were as follows: 94 °C for 10 min; 30 cycles of 94 °C for 1 min, 60 °C for 1 min, 72 °C for 1 min, and a final extension for 5 min at 72 °C.

### 2.11. Molecular Detection of Troglodytella spp.

Detection of *Troglodytella* spp. was only attempted in captive NHP. Identification of this ciliate mutualist was carried out by a direct PCR method targeting a 401 bp fragment of the ITS region of the rDNA (ITS1-5.8S rDNA-ITS2) of the protist [37]. PCR reactions (25 µl) contained 2 µL of template DNA and 0.8 µM of each primer SSU-end/LSU-start (Appendix A). Conditions of PCR for ITS amplification were initial denaturation for 2 min at 94 °C, 35 cycles of 45 s at 94 °C, 45 s at 50 °C, and 90 s at 72 °C, and terminal elongation for 5 min at 72 °C.

### 2.12. Molecular Detection of Leishmania spp.

Detection of *Leishmania* spp. was solely attempted in rodents, the only mammalian host for which tissue samples were available. Identification of this kinetoplastida parasite was carried out by a nested PCR protocol to amplify a partial fragment (358 bp) of the *ssu* rRNA gene of the parasite [38]. The primary PCR reaction (50 µL) contained 10 µL of template DNA and 15 pmol of the primer pair R221/R332 (Appendix A). Conditions of PCR for *ssu* rRNA amplification were initial denaturation for 5 min at 94 °C, 35 cycles of 30 s at 94 °C, 30 s at 60 °C, and 30 s at 72 °C, and terminal elongation for 10 min at 72 °C. In the secondary PCR reaction (25 µL), 10 µL of a 1:40 dilution of the primary PCR product was re-amplified using 7.5 pmol of the primer pair R223/R333 (Appendix A). Cycling conditions were as described above except that the annealing temperature was set at 65 °C.

All the direct, semi-nested, and nested PCR protocols described above were conducted on a 2720 Thermal Cycler (Applied Biosystems). Reaction mixes always included 2.5 units of MyTAQ^TM^ DNA polymerase (Bioline GmbH, Luckenwalde, Germany), and 5× MyTAQ^TM^ Reaction Buffer containing 5 mM dNTPs and 15 mM MgCl_2_, except for the amplification of *Leishmania* spp., for which 0.7–1.4 units of Tth DNA polymerase (Biotools B&M Laboratories, S.A., Madrid, Spain) were used. Laboratory-confirmed positive and negative DNA samples of human and animal origin for each parasitic species investigated were routinely used as controls and included in each round of PCR. PCR amplicons were visualized on 1.5–2% D5 agarose gels (Conda, Madrid, Spain) stained with Pronasafe (Conda) or Gel Red (Biotium, Fremont, CA, USA) nucleic acid staining solutions. A 100 bp DNA ladder (Boehringer Mannheim GmbH, Baden-Wurttemberg, Germany) was used for the sizing of obtained amplicons. Positive-PCR products were directly sequenced in both directions using appropriate internal primer sets (Appendix A). DNA sequencing was conducted by capillary electrophoresis using the BigDye^®^ Terminator chemistry (Applied Biosystems) on an on ABI PRISM 3130 automated DNA sequencer.

The sequences obtained in this study have been deposited in GenBank under accession numbers MW417420–MW417422 (*G. duodenalis*), MW414634–MW414644 and MW581486 (*Blastocystis* sp.), MW406908–MW406921 (*Cryptosporidium* spp.) and MW414645 (*E. bieneusi*).

### 2.13. Statistical Analysis

Prevalence and 95% confidence intervals (95% CI) of any enteric protist infection/carriage, alone or in combination, in the study populations were calculated. Statistically significant differences between the prevalence of enteric protist species in NHP and sampling period were analyzed using the Pearson’s chi-square or Fisher’s exact test with crude odds ratios (OR) and 95% CI, as appropriate. A *p* value < 0.05 was considered evidence of statistical significance. Data were analyzed using R open-source software. Because of the relatively small sample size, limited number of positives obtained from human stool samples, and associated low statistical power, no attempts were conducted to investigate potential correlations between the occurrence of the detected protist species and the risk factors covered in the epidemiological questionnaire provided to volunteer zookeepers.

## 3. Results

### 3.1. Prevalence and Molecular Characterization of Enteroparasites in Captive Non-Human Primates

A total of 51 faecal samples from 10 different species of NHP hosted at the CZCC were collected during the period of study, 28 in the first sampling period and 23 in the second sampling period (Table 1). Members of all 10 NHP species were represented in the two sampling periods. All collected samples could be assigned to individual NHP, except those from the Lemuridae family sharing the same enclosure. Five protist species were detected, including *Blastocystis* sp. (45.1%, 23/51; 95% CI: 31.1–59.7), *E. dispar* (27.5%, 14/51; 95% CI: 15.9–41.7), *G. duodenalis* (21.6%, 11/51; 95% CI: 11.3–35.3), *B. coli* (3.9%, 2/51; 95% CI: 0.5–13.5), and *E. bieneusi* (2.0%, 1/51; 95% CI: 0.05–10.5). In contrast, *Cryptosporidium* spp., *E. histolytica*, and *Troglodytella* spp. were not detected in any of the NHP faecal samples analyzed (Table 1). *Blastocystis* sp. (39.1–50.0%), *E. dispar* (14.3–43.5%), *G. duodenalis* (17.9–26.1%), and *B. coli* (3.6–4.3%) were detected in both sampling campaigns, whereas the only sample that tested positive for *E. bieneusi* was obtained in the second sampling campaign. *Entamoeba dispar* was significantly more prevalent in the second sampling campaign than in the first sampling campaign (χ^2^ = 5.4034, *p* = 0.0201).

Table 2 summarizes the occurrence of the enteric protist species detected in the present survey as single or multiple infections (*n* = 32). Multiple infections with two protist species were found in 14 samples (43.8%, 14/32) of which five (15.6%, 5/32) were co-infected with *Blastocystis* sp. and *E. dispar*, four (12.5%, 4/32) with *Blastocystis* sp. and *G. duodenalis* and three (9.4%, 3/32) with *G. duodenalis* and *E. dispar*. Three samples (9.4%, 3/32) were co-infected with *G. duodenalis*, *Blastocystis* sp., and *E. dispar*. No associations were demonstrated between *Blastocystis* sp. and *E. dispar* (*p* = 0.630; OR = 1.3, 95% CI: 0.41–4.37) or between *Blastocystis* sp. and *G. duodenalis* (*p* = 0.190; OR = 2.6, 95% CI: 0.54–14.1).

*Giardia duodenalis*-positive results by qPCR (*n* = 11) generated cycle threshold (Ct) values ranging from 30.0 to 37.4 (median: 32.4; standard deviation: 2.4). Only a single sample could be genotyped at the *bg* locus, being identified as sub-assemblage AII (Table 3). Sequence alignment analysis revealed that this sequence was identical to its corresponding reference sequence (GenBank accession number: L40510). Out of the 23 *Blastocystis*-positive samples at the *ssu* rDNA (barcode region), the gene of the parasite confirmed by Sanger sequencing revealed the presence of three *Blastocystis* subtypes (STs), including zoonotic ST1 (39.1%, 9/23), ST3 (34.8%, 8/23), and ST8 (26.1%, 6/23) (Table 3). Additionally, 11 samples yielded amplicons of the expected size but in the form of faint bands on gel electrophoresis. Because their associated Sanger sequences were of poor quality (unreadable), these samples were conservatively considered as negative for *Blastocystis* sp. Neither mixed infection involving different STs of the parasite nor infections caused by animal-specific (ST10–ST17, ST21, ST23–ST28) subtypes were identified. A moderate genetic diversity was observed within ST1 (alleles 1 and 2, alone or in combination), and ST3 (alleles 34, 32 + 34), but not within ST8, where all isolates were assigned to allele 21. *Balantioides coli* was unmistakably identified in two isolates, but sequence data of insufficient quality precluded the possibility of determining the genotype of this parasite species. Finally, sequence analysis of the only sample positive to *E. bieneusi* revealed the presence of genotype D with 100% identity with reference sequence AF101200 (Table 3).

### 3.2. Prevalence and Molecular Characterization of Enteroparasites in Humans

A total of 19 members of the CZCC personnel, including zookeepers and veterinarians, participated in the study, 15 of them in the first sampling campaign and 11 in the second sampling campaign. Seven zookeepers participated in both sampling campaigns. The male/female ratio was 3.8, and the age range was 21 to 58 years (median: 49 years). Three individuals tested positive for at least one enteroparasite. Two enteric protist species were identified including *G. duodenalis* (10.5%, 2/19; 95% CI: 1.3–33.1) and *Blastocystis* sp. (10.5%, 2/19; 95% CI: 1.3–33.1). *Giardia duodenalis* was detected by qPCR (Ct values: 30.8 and 31.0) in a 58-year-old male and a 49-year-old female, respectively, participating in the second sampling campaign. Both samples failed to be amplified at the *gdh*, *bg*, and *tpi* loci, so the assemblages/sub-assemblages causing the infections were unknown. *Blastocystis* ST3 allele 34 (GenBank accession number: MW414642) was detected in a 56-year-old male participating in the first sampling campaign, whereas ST1 (GenBank accession number: MW414641) was identified in the same 49-year-old female co-infected with *G. duodenalis*. Sequence analysis of the later isolate revealed two clear double peaks (R and W) at positions 128 and 264, respectively, of reference sequence MK357786, compatible with mixed infections involving alleles 1, 2, 5 and/or 141. The variables potentially associated with *G. duodenalis* infections or *Blastocystis* sp. carriage are summarized in Table 4. The three individuals harbouring *G. duodenalis* and/or *Blastocystis* sp. declared no gastrointestinal symptoms at the moment of sampling. All three were food handlers and were regularly in contact with faecal material from NHP and other captive animal species at the CZCC. Other enteric protist species, including *Cryptosporidium* spp., *E. histolytica*, *E. dispar*, *E. bieneusi*, and *B. coli*, were apparently absent in the surveyed human population.

### 3.3. Prevalence and Molecular Characterization of Enteroparasites and Leishmania spp. in Rats

A total of 64 faecal samples of free-living sympatric rats captured within the premises of the zoological garden were available for this study. Three enteric protist species were detected—*Cryptosporidium* spp. (45.3%, 29/64; 95% CI: 32.8–58.2), *G. duodenalis* (14.1%, 9/64; 95% CI: 6.6–25.0), and *Blastocystis* sp. (6.25%, 4/64; 95% CI: 0.4–10.8). None of the samples tested positive for *E. bieneusi* or *B. coli*. All the spleen, liver, and skin samples analyzed tested negative for *Leishmania* spp.

Sequence analyses of the murine *Cryptosporidium*-positive samples by *ssu*-PCR revealed the presence of *C. muris* (10.3%, 3/29), *C. ratti* (17.2%, 5/29,), rat genotype IV (69.0%, 20/29), and rat genotype V (3.5%, 1/29) (Table 5). All *C. muris* and *C. ratti* showed 100% identity with reference sequences AB089284 and MT504541, respectively. Conversely, a high genetic diversity was found within sequences belonging to rat genotype IV, with only four of them being identical to reference sequence JN172970. The remaining 16 sequences varied from JN172970 by 1–5 single nucleotide polymorphisms (SNPs), including a variety of mutations, insertions, deletions, and ambiguous (double peak) positions, the combination 448DelT + G493A being the most frequently detected (Table 5). The only sequence identified as rat genotype V varied from reference sequence MT504543 by a single (A667G) SNP.

Rodent *G. duodenalis*-positive samples by qPCR generated Ct values ranging from 24.1 to 36.3 (median: 31.8). Of these, 55.6% (5/9) produced Ct values higher than 30. Two *G. duodenalis*-positive samples (Ct values: 24.1 and 28.2, respectively) were genotyped as assemblage G at the *gdh* locus. Both sequences were identical between them and showed an SNP (C262T) compared to reference sequence MF671912. Additionally, one of the sequences was confirmed as assemblage G at the *bg* locus and showed 100% identity with reference sequence MF671912. None of the two isolates could be amplified at the *tpi* locus.

The two *Blastocystis*-positive samples by *ssu*-PCR and Sanger sequencing were identified as ST4 alleles 92 and 94 (GenBank accession numbers: MW414643 and MW414644), and their sequences were identical with reference sequence MF186667 and MN526920, respectively. As in the case of NHP, three additional samples yielded amplicons of the expected size using barcoding primers but with faint bands on gel electrophoresis that did not produce readable sequences. Those three samples were subjected to a PCR to amplify a different region of the *ssu* rRNA gene, and two were found positive and subjected to next-generation amplicon sequencing. Those two samples were identified as ST4 and showed 100% identity with the reference sequence U26177. Because of lack of confirmation by Sanger sequencing or failing to amplify with an additional primer set, one sample was conservatively considered as negative for *Blastocystis* sp.

### 3.4. Molecular-Based Evidence of Zoonotic Transmission

Within NHP, a mangabey (*C. lunulatus*) investigated during the first sampling campaign was found infected with zoonotic *G. duodenalis* assemblage AII. Two zookeepers participating in the second sampling campaign were also positive for *G. duodenalis*, but lack of genotyping data and different sampling intervals precluded the unambiguous demonstration of zoonotic transmission of *G. duodenalis* infection between NHP and their zookeepers (Figure 2).

*Blastocystis* ST1 alleles 1 and 2 (alone or in combination) were consistently detected in mangabeys (*C. lunulatus*), drills (*M. leucophaeus*), and gibbons (*H. leucogenys* Ogilby) along the whole study period, and in a De Brazza´s monkey (*C. neglectus*) during the second sampling campaign. All these NHP species were housed in close proximity to each other within the CZCC premises. These data strongly suggest that these *Blastocystis* genetic variants were well established in the NHP population hosted at the CZCC. Interestingly, a zookeeper participating in the second sampling of the survey carried a genetic variant of *Blastocystis* ST1 compatible with a mixed infection by alleles 1 + 2 (Figure 2). This zookeeper was a food handler and declared regular contact with the faecal material of all NHP in the CZCC. This finding suggests that NHP were acting as a source of *Blastocystis* infection to the zookeepers responsible for their wellbeing.

Finally, the surveyed rat population was exclusively infected by rodent-specific species/genotypes of *Cryptosporidium*, *G. duodenalis*, and *Blastocystis* (Figure 2), indicating that this host species has a limited role as a source of potential infections for NHP and humans. Additionally, all tested rats were negative to zoonotic *Leishmania* spp.

## 4. Discussion

The epidemiology of pathogenic and commensal enteric protists in captive NHP is poorly understood. To fill this gap of knowledge, this survey provides new molecular-based data on the occurrence, transmission, genetic diversity, and zoonotic potential of the protist species that are most relevant from the public health point of view in NHP, their zookeepers/veterinarians, and free-living rats at the CZCC. In addition, the potential role of rodents as a natural reservoir of *Leishmania* spp. has been investigated.

*Cryptosporidium* spp. (particularly *C. hominis* and *C. parvum*) is, together with rotavirus, *Shigella*, and enterotoxigenic *Escherichia coli*, the major contributors to the global burden of diarrhoeal disease [40]. *Cryptosporidium* spp. is also a common diarrhoea-causing agent in livestock, companion species, and wildlife [6]. Interestingly, *Cryptosporidium* spp. was absent in the NHP population surveyed here and in the zookeepers/veterinarians that worked in their well care. This agrees with the previous findings observed in NHP (*n* = 18) from the Almuñecar zoological garden in southern Spain [17] but is in sharp contrast with those from the Barcelona zoological garden, where *Cryptosporidium* infections were consistently reported in 28–44% of the NHP investigated during a 10-year period [13,15,16]. In those studies, infected NHP were asymptomatic adults with intermittent (up to 10 months) shedding of oocysts irrespectively of the group or lone condition of the animals. This fact suggested that *Cryptosporidium* reinfection rather than continuous infection was taking part in that setting [17]. In contrast, *Cryptosporidium* spp. was found at a high prevalence rate (45%) in free-living rats captured within the CZCC enclosure. In the only two previous studies published in Spain, *Cryptosporidium* spp. infections have been reported in black rats from Catalonia (1/1) and the Canary Islands (14/101) [41,42]. Our sequence analyses revealed the presence of four distinct *Cryptosporidium* species/genotypes including rat genotype IV (69%), *C. ratti* (17%), *C. muris* (10%), and rat genotype V (4%). It should be noted that *C. ratti*, formerly known as rat genotype I, has been recently proposed as a valid *Cryptosporidium* species by Martin Kváč’s laboratory [39]. Of these, *C. muris* and *C. ratti* (in addition to *C. meleagridis* and rat genotype II/III, not identified in the present survey) have been previously described in black rats from the Canary Islands [42]. Overall, our data indicate that rats captured at the CZCC were infected by murine-adapted *Cryptosporidium* species/genotypes and played a limited role as a source of cryptosporidiosis to NHP. Of interest, *C. muris* and rat genotype III have been sporadically reported in humans and/or companion animals, including dogs and cats [43,44,45,46].

In the present study, *G. duodenalis* infections were identified in 22% of NHP and 11% of zookeepers. Interestingly, all *Giardia*-positive cases by qPCR yielded Ct values >30, indicative of moderate-to-low parasite burdens. This agrees with the fact that all positive cases were asymptomatic and produced formed stools, also explaining the low genotyping success rate obtained (7.7%, 1/13). It should be noted that *gdh*, *bg*, and *tpi* are all single-copy genes with limited sensitivity compared with the multiple-copy *ssu* rRNA gene used in qPCR for detection purposes. An early epidemiological study detected the presence of *G. duodenalis* in 19.1% of NHP in the Barcelona zoological garden [14], but this parasite was absent in the NHP analyzed at the Almuñecar zoological garden [17]. Our sequence analyses identified the zoonotic sub-assemblage AII in a mangabey (*C. lunulatus*). In Spain, the *G. duodenalis* sub-assemblage AII has been found in 15–44% of documented clinical cases [47,48], and in 17–33% of children of paediatric age [49,50]. Although two of the CZCC zookeepers tested positive to this protozoan parasite, we were unable genotype these isolates, so their assemblage/sub-assemblage remained unknown and precluded us to propose a potential source of infection. Of interest, zoonotic AI and BIV have been previously identified in members of the Lemuridae family in the Valencia and Madrid zoological gardens [18].

*Blastocystis* infection/carriage was demonstrated in 45.1% of the NHP surveyed, a frequency rate considerably lower than those (67–95%) previously reported by conventional microscopy at the Barcelona and Almuñecar zoological gardens, respectively [14,17]. Sequence analyses of *Blastocystis* isolates revealed interesting data. ST1 was the most prevalent (39%) subtype circulating among captive NHP, being present in two genetic variants, allele 1 and allele 2, either alone or in combination. We have recently reported ST1 as the most common (82%) *Blastocystis* subtype in wild western chimpanzees (*Pan troglodytes verus*) in Senegal, although in that survey all the isolates characterized belonged to alleles 7 and 8 [51]. Moreover, we have also demonstrated that human cases of blastocystosis by ST1 in Spain are mainly due to allele 4 (and, to a much lesser extent, allele 77) both in asymptomatic [50,52] and clinical [53] individuals. The fact that one of the CZCC primate handlers carried a genetic variant of *Blastocystis* ST1 compatible with a mixed infection involving alleles 1, 2, 5 and/or 141 seems to indicate that this ST1 infection is most likely of primate origin and represents a zoonotic transmission event. Similarly, the vast majority of the ST3 isolates detected in NHP at the CZCC belonged to allele 32, a genetic variant not yet described in Spanish human populations [50,52,53]. This fact may indicate that ST3 allele 32 may be better adapted to infect NHP than humans. Finally, *Blastocystis* ST8 carriage was also a common finding (26%) in NHP at the CZCC. This result was highly expected as this *Blastocystis* subtype is well-known both in captive [54] and free-living [55] NHP globally. Although rarely reported in humans, the zoonotic potential of ST8 has been demonstrated in a zoological garden in the UK, where this subtype was responsible from one in four *Blastocystis* infections both in captive NHP and their handlers [56]. Also relevant was the finding of only identifying *Blastocystis* ST4 in rats. This finding supports that rodents appear to constitute the main animal reservoir of ST4 [56,57]. Additionally, the marked geographical distribution of ST4 in humans (commonly found in Europe but rarely or less frequently present in other geographical areas), together with its clonal structure strongly suggest that ST4 represents a lineage with a recent entry into the human population [58]. In Spain, all human cases carrying *Blastocystis* ST4 have been assigned to the allele 42 of the protist. As in the case of ST3 allele 32, this fact may indicate that ST4 alleles 92 and 94 (identified in rats in this study) may be particularly adapted to infect/colonize rodent species rather than humans.

*Enterocytozoon bieneusi* genotype D was detected in a single gibbon (*H. leucogenys* Ogilby). Genotype D has broad host and geographic ranges and belongs to Group 1 that includes zoonotic *E. bieneusi* genotypes most frequently found in humans, domestic and wild (including NHP) animal species worldwide [59,60]. In Spain, *E. bieneusi* genotype D has been described in renal transplant recipients [61], domestic rabbits [62] and cats [63], wild red foxes [62], and environmental (water) samples [64]. This result clearly indicates that NHP may act as suitable reservoirs for human microsporidiosis by *E. bieneusi*.

Regarding ciliate species, zoonotic *B. coli* was identified in a low (3.9%) proportion of the NHP investigated, but not in their handlers. At first sight, this result is much lower than those previously documented by microscopy examination in NHP at the Barcelona (38.1%) and Almuñecar (16.6%) zoological gardens [14,17]. However, when considering primate groups, *B. coli* has been found only in Catarrhini (*Mandrillus*, present results; *Cercocebus*, *Gorilla*, *Pan*, *Papio*, and *Pongo*) [14]. The data from Pérez-Cordón et al. [17] are aggregated and it is not possible to identify the positive primate genera. Our negative results in Strepsirrhini (*Eulemur*, *Lemur*, and *Varecia*) and Plathyrrhini (*Saimiri*) primates are in accordance with previous data [19] and suggest that these primates are uncommon or not valid hosts for this ciliate. It should be noted that, because *B. coli* cysts are morphologically indistinguishable from other ciliate species (i.e., *Buxtonella* spp.), it is possible that some of the microscopy-based prevalence rates described above do indeed represent an overestimation of the true occurrence of the parasite. While *B. coli*-like cysts are easily identifiable by microscopy, differential diagnosis based on molecular (PCR and Sanger sequencing) methods should be used for the correct identification of *B. coli*.

Finally, the absence of *T. abrassarti* in the investigated species is in accordance with previous surveys [65]. This ciliate is commonly reported in wild great apes [20] but there is no conclusive evidence in lesser apes and monkeys (e.g., red colobus, red-tailed monkeys, vervet monkeys, and yellow baboons). In captive chimpanzees, prevalence and infection intensities are influenced by the dietary starch concentration, suggesting a symbiotic function and participation in nourishment degradation [66].

Vector-borne *Leishmania infantum*, the causative agent of visceral and cutaneous leishmaniasis in Spain, is one of the most important neglected zoonosis in the Mediterranean region. In Spain, and in addition to domestic dogs, leporids such as rabbits and hares have been demonstrated as competent reservoirs of the infection [67]. Micromammals (e.g., mice, shrews) seem to play a limited role in the epidemiology of the parasite [68], although an unanticipated high prevalence rate (33%) of the parasite has been recently described in rats captured in sewers in the city of Barcelona [69]. To confirm the accuracy and extent of these previous findings, we investigated by PCR the occurrence of *Leishmania* spp. in rat tissues, including liver, spleen, and ear skin. In all cases, we failed to detect the presence of the parasite. It is possible that the limited number of rodent samples (*n* = 64) analyzed in the present study may have biased the obtained results. Further studies are warranted to assess the role of rodent on *Leishmania* spp. transmission in these epidemiological scenarios.

## 5. Conclusions

A high prevalence of the diarrhoea-causing protists *G. duodenalis* and *Blastocystis* sp. (but not *Cryptosporidium* spp.) was observed in captive NHP at the CZCC. NHP can harbour zoonotic genotypes of *G. duodenalis*, *Blastocystis* sp., and *E. bieneusi*. Indeed, strong evidence of the occurrence of *Blastocystis* zoonotic transmission between NHP and their handlers was provided, despite the use of personal protective equipment and the implementation of strict health and safety protocols. Free-living sympatric rats are infected by host-specific species/genotypes of the investigated protists and seem to play a limited role as a source of infections to NHP or humans in this setting. The extent of these findings should be confirmed in similar epidemiological surveys targeting other captive NHP populations.

## Figures and Tables

**Figure 1 animals-11-00700-f001:**
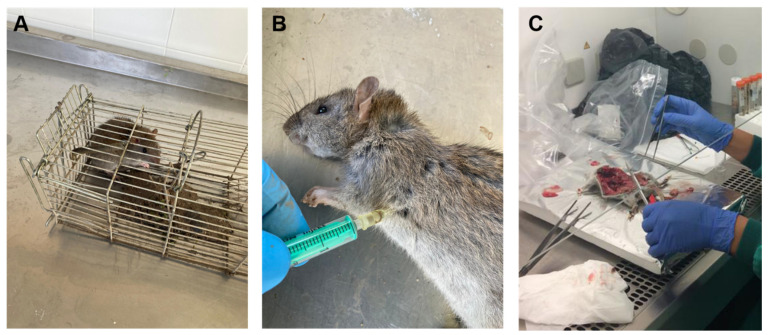
Sampling of rodent specimens within the premises of the Córdoba Zoo Conservation Centre. (**A**): Capture using live traps; (**B**): Humanely killing by intracardiac injection of sodium pentobarbitone; (**C**): Dissection of rat carcasses and organ removal.

**Figure 2 animals-11-00700-f002:**
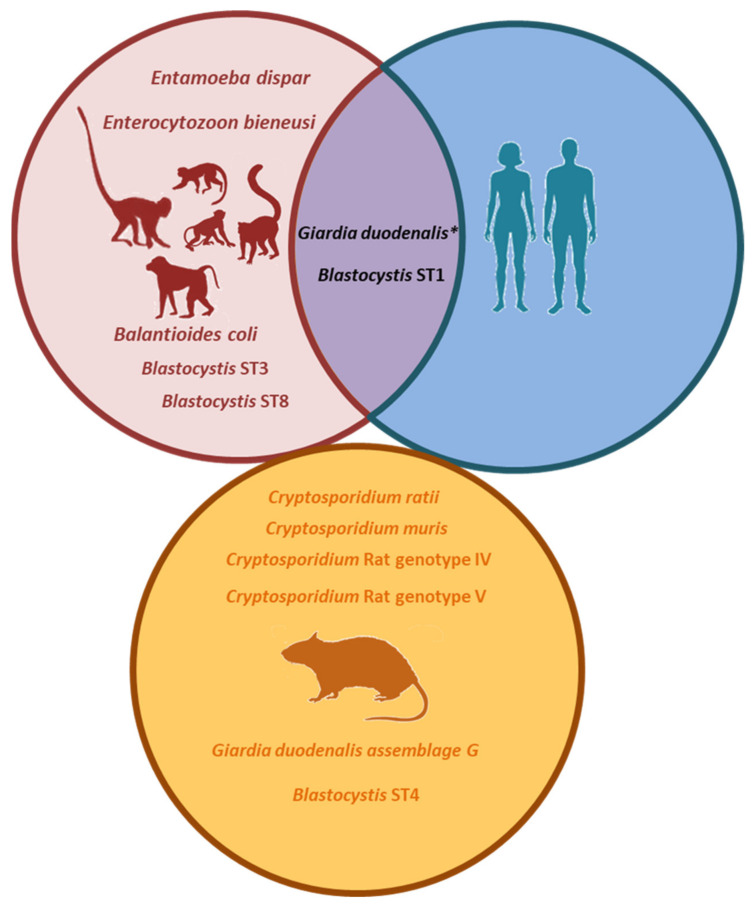
Molecular-based evidence of zoonotic transmission. Enteric protists detected at each species group (non-human primates (NHP), zookeepers and rats) in the Córdoba Zoo Conservation Centre (Spain). * Zoonotic *Giardia duodenalis* assemblage AII was detected in NHP; in addition, two zookeepers were positive for *G. duodenalis* of unknown assemblage, potentially making the assessment of zoonotic transmission for this protozoan parasite difficult.

**Table 1 animals-11-00700-t001:** Frequency of enteric protists detected at each sampling campaign in faecal samples from captive non-human primates in the Córdoba Zoo Conservation Centre (Spain).

	First Sampling Campaign						Second Sampling Campaign						All					
		Frequency Positive Results (%)						Frequency Positive Results (%)						Frequency Positive Results (%)				
Species	No.	Bl	Ed	Gd	Bc	Eb	No.	Bl	Ed	Gd	Bc	Eb	No.	Bl	Ed	Gd	Bc	Eb
*Cebuella pygmaea*	1	0.0	0.0	0.0	0.0	0.0	2	0.0	0.0	50.0	0.0	0.0	3	0.0	0.0	33.3	0.0	0.0
*Cercocebus lunulatus*	3	100	0.0	100	0.0	0.0	3	33.3	100	66.7	0.0	0.0	6	66.7	50.0	83.3	0.0	0.0
*Cercopithecus neglectus*	2	50.0	0.0	50.0	0.0	0.0	3	66.7	66.7	0.0	0.0	0.0	5	60.0	40.0	20.0	0.0	0.0
*Eulemur fulvus*	2	0.0	0.0	0.0	0.0	0.0	0	0.0	0.0	0.0	0.0	0.0	2	0.0	0.0	0.0	0.0	0.0
*Hylobates leucogenys* Ogilby	4	75.0	100	25.0	0.0	0.0	3	33.3	33.3	0.0	0.0	33.3	7	57.1	71.4	14.3	0.0	14.3
*Lemur catta*	2	100	0.0	0.0	0.0	0.0	0	0.0	0.0	0.0	0.0	0.0	2	100	0.0	0.0	0.0	0.0
*Macaca sylvanus*	5	40.0	0.0	0.0	0.0	0.0	3	33.3	100	100	0.0	0.0	8	37.5	37.5	37.5	0.0	0.0
*Mandrillus leucophaeus*	5	40.0	0.0	0.0	20.0	0.0	3	33.3	33.3	0.0	33.3	0.0	8	37.5	12.5	0.0	25.0	0.0
*Saimiri sciureus*	2	0.0	0.0	0.0	0.0	0.0	3	0.0	0.0	0.0	0.0	0.0	5	0.0	0.0	0.0	0.0	0.0
*Varecia variegata variegata*	2	50.0	0.0	0.0	0.0	0.0	3	100	0.0	0.0	0.0	0.0	5	80.0	0.0	0.0	0.0	0.0
Total	28	50.0	14.3	17.9	3.6	0.0	23	39.1	43.5	26.1	4.3	4.3	51	45.1	27.5	21.6	3.9	2.0

Bc: *Balantioides coli*; Bl: *Blastocystis* sp., Eb: *Enterocytozoon bieneusi*; Ed: *Entamoeba dispar*; Gd: *Giardia duodenalis*.

**Table 2 animals-11-00700-t002:** Single and multiple enteric protist infections detected in faecal samples from captive non-human primates in the Córdoba Zoo Conservation Centre (Spain).

Species Combination	No. of Faecal Samples
*Blastocystis* sp. Only	10
*E. dispar* only	4
*G. duodenalis* only	1
*Blastocystis* sp. + *E. dispar*	5
*Blastocystis* sp. + *G. duodenalis*	4
*G. duodenalis* + *E. dispar*	3
*Blastocystis* sp. + *B. coli*	1
*E. dispar* + *E. bieneusi*	1
*G. duodenalis* + *Blastocystis* sp. + *E. dispar*	3
Total	32

**Table 3 animals-11-00700-t003:** Diversity, frequency, and molecular features of *Giardia duodenalis*, *Blastocystis* sp., *Balantioides coli,* and *Enterocytozoon bieneusi* in faecal samples from captive non-human primates in the Córdoba Zoo Conservation Centre (Spain). GenBank accession numbers are provided.

Species	Genotype	Sub-Genotype	Host Species	No. of Isolates	Locus	Reference Sequence	Stretch	Single Nucleotide Polymorphisms	GenBank ID
*Giardia duodenalis*	*A*	AII	*C.t.*	1	*bg*	AY072723	205–539	None	MW417420
*Blastocystis sp.*	ST1	Allele 1	*M.l.*, *H.l.*, *C.t.*	4	*ssu* rRNA	MK357786	4–602	None	MW414634
		Allele 2	*C.t.*	1	*ssu* rRNA	MT094302	36–539	None	MW414635
		Allele 2	*C.n.*	1	*ssu* rRNA	MT094302	32–539	C57A, 65InsG, A112G, C128A, C237T, C272T, A458C	MW414636
		Alleles 1 + 2	*C.t.*, *M.l.*	3	*ssu* rRNA	MK357786	1–603	G128R, A474W	MW414637
	ST3	Allele 34	*H.l.*, *M.c., C.n.*	7	*ssu* rRNA	MK801359	1–581	G114A, A115T, A116G, A159G, T160A, A161T	MW414638
		Alleles 32 + 34	*C.n.*	1	*ssu* rRNA	MK801359	1–586	G114A, A115T, A116G, A159K, T160R, A161K, A162R	MW414639
	ST8	Allele 21	*V.v.v.*, *L. c.*	6	*ssu* rRNA	MT509451	1–525	None	MW414640
*Balantioides coli*	Unknown	-	*M.l.*	2	ITS	-	-	-	-
*Enterocytozoon bieneusi*	D	-	*H.l.*	1	ITS	AF101200	31–419	None	MW414645

bg: β-giardin; *C.n.*: *Cercopithecus neglectus*; *C.t.*: *Cercocebus torquatus*; *H.l.*: *Hylobates leucogenys*; ITS: Internal transcribed spacer; *L.c.*: *Lemur catta*; *M.c.*: *Macaca sylvanus*; *M.l.*: *Mandrillus leucophaeus*; *ssu* rRNA: Small subunit ribosomal RNA; *V.v.v.*: *Varecia variegata variegata*.

**Table 4 animals-11-00700-t004:** Variables potentially associated to *G. duodenalis* infection and *Blastocystis* sp. carriage in staff at the Córdoba Zoo Conservation Centre (Spain).

Variable	Subject 38	Subject 79	Subject 86
**Sociodemographic factors**			
Sex	Male	Male	Female
Age (years)	56	58	49
**Protist infection/carriage**			
*Giardia duodenalis*	Negative	Positive	Positive
*Blastocystis* sp.	Positive	Negative	Positive
**Clinical factors**			
Diarrhoea in the last 7 days	No	No	No
Contact with children <5-years	No	No	No
Diarrhoea in family members/relatives	Yes	No	No
**Work-related factors**			
Activity	Veterinarian	Zookeeper	Zookeeper
Exposure to faeces from NHP	Yes	Yes	Yes
Exposure to faeces from animals other than NHP	Yes	Yes	Yes
Any of these animal species with diarrhoea	Yes	Yes	Yes
Food handler	Yes	Yes	Yes
**Behavioural factors**			
Recent travel	Yes	No	No
Contact with pet dogs	Yes	Yes	Yes
Contact with pet cats	Yes	No	Yes
Main drinking source—tap	Yes	Yes	Yes
Main drinking source—bottled	No	No	No
Swimming	No	No	No
Handwashing	Frequently	Always	Always
Vegetable washing	Always	Always	Always

**Table 5 animals-11-00700-t005:** Diversity, frequency, and molecular features of *Cryptosporidium* spp. sequences at the *ssu* rRNA locus obtained in faecal samples from the rat population under study in Córdoba Zoo Conservation Centre (Spain). GenBank accession numbers are provided.

Species/Genotype	No. of Isolates	Reference Sequence	Stretch	Single Nucleotide Polymorphisms	GenBank ID
*C. muris*	3	AB089284	504–1012	None	MW406908
*C. ratti* ^a^	6	MT504541	293–751	None	MW406909
Rat genotype IV	3	JN172970	377–775	None	MW406910
	1	JN172970	332–798	C342T, C410T, C423T, 448DelT, G493A	MW406911
	1	JN172970	332–752	C342T, C410T, C423T, 490_491DelTT, G493A	MW406912
	1	JN172970	328–814	A428G, 448DelT, G493A	MW406913
	1	JN172970	328–767	A445G, 448DelT, G493A, T541A, T542A	MW406914
	1	JN172970	348–776	A445G, 448DelT, G493A, T541W, T542W	MW406915
	5	JN172970	375–814	448DelT, G493A	MW406916
	1	JN172970	454–678	A459T, A475T, G493A	MW406917
	1	JN172970	332–788	A472G, 492InsT, G493A	MW406918
	3	JN172970	341–814	490_491DelTT, G493A	MW406919
	1	JN172970	481–814	490_491DelTT, G493A, G635A	MW406920
Rat genotype V	1	MT504543	306–699	A667G	MW406921

^a^ Formerly known as Rat genotype I. See reference [39].

## Data Availability

All relevant data are within the article and its additional files. The sequences data were submitted to the GenBank database under the accession numbers MW417420–MW417422 (*G. duodenalis*), MW414634–MW414644 (*Blastocystis* sp.), MW406908–MW406921 (*Cryptosporidium* spp.) and MW414645 (*E. bieneusi*).

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
