# Peer review of "Occurrence and Genetic Diversity of Protist Parasites in Captive Non-Human Primates, Zookeepers, and Free-Living Sympatric Rats in the Córdoba Zoo Conservation Centre, Southern Spain"

_animals, 2021, doi:10.3390/ani11030700_

Round 1

Reviewer 1 Report

The manuscript entitled “Occurrence and genetic diversity of protist parasites in captive non-human primates, zookeepers, and free-living sympatric rats in the Córdoba Zoo Conservation Centre, southern Spain” presents the results of the investigation by molecular methods of infections by 6 enteric protists in three groups of hosts (NHP, humans and rodents) and additionally of the protozoan parasite Leishmania spp. in the rodents only.

To my opinion, this is a very well conducted and presented survey. The methods were well applied and described and the results are comprehensive and well presented. Finally, the discussion is well organised, touches the most important issues and analyses the relevance of the results to the appropriate level.

Overall, I consider this to be an excellent work and suitable for publication and I would only call the attention of the authors to the following minor suggestions/corrections:

Abstract: I believe that the examination and negative results for Troglodytella abrassarti and the result for Leishmania examination in rats should also be mentioned in the Abstract. The information about the human Giardia genotype is also missing, so I believe it would be of benefit to mention that genotyping the human isolated was not successful.

Figure 1C. The tube with the blood is somehow misleading as no examinations in blood samples are reported. Cropping the photograph or using another one would be preferable.

Results, lines 308-310: this is obviously left from the template and should be deleted.

Results, line 408: Please be more specific by writing Rattus spp. in a bracket unless the examination included other rodents too, e.g. Mus spp. so in that case please change "rats" to "small rodents".

Discussion line 488: Please add “diarrhoea” after “agent”

Discussion lines 508-511: Before this concluding sentence it would be helpful to write a sentence about the host specificity of the identified genotypes, supported by some references that certify that these are considered host-specific.

Author Response

Referee #1

The manuscript entitled “Occurrence and genetic diversity of protist parasites in captive non-human primates, zookeepers, and free-living sympatric rats in the Córdoba Zoo Conservation Centre, southern Spain” presents the results of the investigation by molecular methods of infections by 6 enteric protists in three groups of hosts (NHP, humans and rodents) and additionally of the protozoan parasite Leishmania spp. in the rodents only. To my opinion, this is a very well conducted and presented survey. The methods were well applied and described and the results are comprehensive and well presented. Finally, the discussion is well organised, touches the most important issues and analyses the relevance of the results to the appropriate level. Overall, I consider this to be an excellent work and suitable for publication and I would only call the attention of the authors to the following minor suggestions/corrections:

We thank Referee #1 for his/her preliminary positive assessment.

  1. Abstract: I believe that the examination and negative results for Troglodytella abrassarti and the result for Leishmania examination in rats should also be mentioned in the Abstract. The information about the human Giardia genotype is also missing, so I believe it would be of benefit to mention that genotyping the human isolated was not successful.

Reply: Please note that, following the Authors Guidelines, the abstract of the paper should have a maximum length of 200 words. We have attempted to accommodate the information requested by Referee #1 in current lines 51, 53, and 55, but consequently the word count has now increased to 234.

  1. Figure 1C. The tube with the blood is somehow misleading as no examinations in blood samples are reported. Cropping the photograph or using another one would be preferable.

Reply: Figure 1C has been now replaced by a picture illustrating the dissection of rat carcasses and the extraction of the organs of interest. The Figure title has been modified accordingly.

  1. Results, lines 308-310: this is obviously left from the template and should be deleted.

Reply: We thank Referee #1 for spotting this and apologize for the mistake. The above-mentioned paragraph has been removed from the text.

  1. Results, line 408: Please be more specific by writing Rattus in a bracket unless the examination included other rodents too, e.g. Mus spp. so in that case please change "rats" to "small rodents".

Reply: Referee #1 raised an important point. Most captured rats were identified as brown rats (Rattus norvegicus), but differential diagnosis based only on morphological features with black rats (Rattus rattus) was not possible for younger individuals. This issue has been now acknowledged in lines 142-144 of the Materials and Methods section.

  1. Discussion line 488: Please add “diarrhoea” after “agent”

Reply: The expression has been now modified as “diarrhoea-causing agent” in current lines 503-504.

  1. Discussion lines 508-511: Before this concluding sentence it would be helpful to write a sentence about the host specificity of the identified genotypes, supported by some references that certify that these are considered host-specific.

Reply: To avoid confusion and misleading interpretations the term “host-specific” has been replaced by the term “murine-adapted” in line 525 of the Discussion section. In this way we provide a justification for the occasional finding of e.g., C. muris infections in humans and carnivores reported in the literature. To back up this statement we have now added four relevant references (Guy et al., Parasit Vectors. 2021;14:69; Ayinmode et al. Vet Parasitol Reg Stud Reports. 2018;14:54-58; Yang et al. Exp Parasitol. 2015 Aug;155:13-8, and Pavlasek and Ryan, Vet Parasitol. 2007;144:349-52) in current lines 526-528. Subsequent reference numbering has been modified accordingly.

Reviewer 2 Report

The manuscript "Occurrence and genetic diversity of protist parasites in captive non-human primates, zookeepers, and free-living sympatric rats in the Córdoba Zoo Conservation Centre, southern Spain" concerns a study of molecular epidemiology on selected potentially zoonotic protozoa and protists in non-human primates kept in a zoological garden in southern Spain, in some keepers and veterinarians, and in synanthropic rodents captured within the same zoo. The study is very interesting, aimed at investigating the dynamics of transmission of infections between animals and humans. The manuscript is well written, the methodology used clearly described and the results well organized and discussed.

I therefore recommend accepting the manuscript, after minor revisions, as reported below.

- line 84, “At least 28 subtypes (ST)”: I would rather say that 24 STs are currently considered valid

- lines 116-118: it may be useful to add the total number of NHPs hosted in the CZCC

- lines 128-130: are they individual samples, it was possible to attribute each sample to a particular individual?

- line 177: I suggest adding how much time elapsed between the DNA extraction and the molecular analyzes

- line 194: I suggest adding a description of the PCR conditions, as done for the other protocols

- line 211: why were two different protocols used for NHP / humans and rats?

- line 297: I suggest adding that co-infections have also been evaluated

- lines 308-310: delete “This section may be divided by subheadings. It should provide a concise and precise description of the experimental results, their interpretation, as well as the experimental conclusions that can be drawn.”

- lines 337-344: I think this paragraph is of little use, considering that this information is contained in Table 1

- lines 358-359: of the 11 samples analyzed on the three different genes used for genotyping, was it possible to amplify only the bg of a single sample? I suggest specifying this data. I suggest also adding in the sentence to which host species the sequenced sample belongs

- Table 3: I suggest adding in the table the hosts in which each pathogen has been found

- Line 386: I would add here that in total 3 people tested positive for at least one of the pathogens investigated. Is it possible to know if they are veterinarians or keepers?

- line 397, “risk factors”: I would not call them risk factors, as no statistical analysis was carried out. I would call them, as in Table 4, "variables"

- LINE 408, “RATS”: Has species identification been carried out? if this data is available, I think it is important to add it

- line 475: Blastocystis has been identified as St4 (I read it from figure 2) but which allele has been identified?

- Line 484, “zookeepers”: and veteriarians. Check elsewhere (e.g. line 490)

- line 486: as Cryptosporidium has not been found in NHPs and humans, I suggest moving this paragraph after the Giardia and Blastocystis paragraphs.

- line 545: any information about other countries?

- line 553: as written above, I would specify the identified allele(s)

- line 554: you can also refer to other bibliography, check e.g. Mohammadpour et al., 2020, Parasites and Vectors 13, Issue 1, Article number 365

- Lines 558-559: Please add a reference. Also, consider that elsewhere other ST4 alleles have been found in humans (see e.g. Jimenez et al., 2019 Parasites & Vectors volume 12, Article number: 376 )

Author Response

Referee #2

The manuscript "Occurrence and genetic diversity of protist parasites in captive non-human primates, zookeepers, and free-living sympatric rats in the Córdoba Zoo Conservation Centre, southern Spain" concerns a study of molecular epidemiology on selected potentially zoonotic protozoa and protists in non-human primates kept in a zoological garden in southern Spain, in some keepers and veterinarians, and in synanthropic rodents captured within the same zoo. The study is very interesting, aimed at investigating the dynamics of transmission of infections between animals and humans. The manuscript is well written, the methodology used clearly described and the results well organized and discussed. I therefore recommend accepting the manuscript, after minor revisions, as reported below.

We thank Referee #1 for his/her preliminary positive assessment.

  1. line 84, “At least 28 subtypes (ST)”: I would rather say that 24 STs are currently considered valid.

Reply: in order to avoid confusion whe have add the following sentence in the paragraph: “A recent evaluation of ST1-ST26 subtypes concluded that only 22 of those subtypes (ST1-ST17, ST21, ST23-ST26) should be acknowledged as legitimate subtypes [10], with the remaining six pending of confirmation in future investigations”.

  1. lines 116-118: it may be useful to add the total number of NHPs hosted in the CZCC

Reply: the total population of NHP hosted at the CZCC was n =36. The number of individuals per genera has been now specified in the Materials and methods section of the manuscript.

  1. lines 128-130: are they individual samples, it was possible to attribute each sample to a particular individual?

Reply: Yes, we were able to link most of the collected faecal samples to a particular individual. Please note that this information was already mentioned in lines 336-337 of the manuscript, where we stated that “All collected samples could be assigned to individual NHP, except those from the Lemuridae family sharing the same enclosure”.

  1. line 177: I suggest adding how much time elapsed between the DNA extraction and the molecular analyses

Reply: All PCR testing was conducted in the two weeks following DNA extraction and purification. DNA samples were then stored at -20 °C. This is a standard procedure in clinical and research laboratories so in our opnion there is no need to state it in the main body of the manuscript.

  1. line 194: I suggest adding a description of the PCR conditions, as done for the other protocols

Reply: A short description of the PCR protocols used for the genotyping of G. duodenalis isolates at the gdh, bg, and tpi loci has been now provided in lines 198 to 217 of the Materials and Methods section.

  1. line 211: why were two different protocols used for NHP / humans and rats?

Reply: NGS was used in samples of murine origin for which Sanger sequencing results were unclear or of insufficient quality. This point has been now highlighted in current lines 233 and 234 of the manuscript.

  1. line 297: I suggest adding that co-infections have also been evaluated.

Reply: Added as per requested in current line 320 of the Materials and methods section.

  1. lines 308-310: delete “This section may be divided by subheadings. It should provide a concise and precise description of the experimental results, their interpretation, as well as the experimental conclusions that can be drawn.”

Reply: We thank Referee #2 for spotting this and apologize for the mistake. The above-mentioned paragraph has been removed from the text.

  1. lines 337-344: I think this paragraph is of little use, considering that this information is contained in Table 1

Reply: Following Referee #2 advice, this paragraph has been now removed from the main body of the text.

  1. lines 358-359: of the 11 samples analyzed on the three different genes used for genotyping, was it possible to amplify only the bg of a single sample? I suggest specifying this data. I suggest also adding in the sentence to which host species the sequenced sample belongs

Reply: please note that in the present survey detection of G. duodenalis was accomplished by qPCR targeting the multiple-copy ssu rRNA gene, far more sensitive than the single copy genes (gdh, bg, and tpi) used here for genotyping purposes. In practical terms this means that samples with qPCR Ct values higher than 30 are unlikely to be successfully amplified by gdh-PCR, bg-PCR or tpi-PCR. This limitation of the study was already acknowledged in current lines533-535 of the Discussion section.

  1. Table 3: I suggest adding in the table the hosts in which each pathogen has been found

Reply: Host species has been added in Table 3 according to Referee #2 suggestion. The Table foot has been modified accordingly.

  1. Line 386: I would add here that in total 3 people tested positive for at least one of the pathogens investigated. Is it possible to know if they are veterinarians or keepers?

Reply: The requested information has been now added in current line 399 and in Table 4.

  1. line 397, “risk factors”: I would not call them risk factors, as no statistical analysis was carried out. I would call them, as in Table 4, "variables"

Reply: Following Referee #2 suggestion the term “risk factor” has been replaced by the term “variable”.

  1. LINE 408, “RATS”: Has species identification been carried out? if this data is available, I think it is important to add it

Reply: Please see our reply to comment #4 by Referee #1.

  1. line 475: Blastocystis has been identified as ST4 (I read it from figure 2) but which allele has been identified?

Reply: Please note that this information was already stated in current lines 451-452 (“The two Blastocystis-positive samples by ssu-PCR were identified as ST4 alleles 92 and 94…” and 458-459 (“Those two samples were identified as ST4 and…”) of the manuscript.

  1. Line 484, “zookeepers”: and veteriarians. Check elsewhere (e.g. line 490)

Reply: The term “zookeepers/veterinarians” has been now introduced in current lines 498 and 506 of the manuscript.

  1. line 486: as Cryptosporidium has not been found in NHPs and humans, I suggest moving this paragraph after the Giardia and Blastocystis

Reply: Please note that the order of the protist species in the Discussion section obey to their veterinary health relevance. For this reason, we prefer to mention first Cryptosporidium spp. over the remaining protist species investigated here.

  1. line 545: any information about other countries?

Reply: No to the best of our knowledge. Please note that allele calling is far less frequently reported in the literature than Blastocystis STs.

  1. line 553: as written above, I would specify the identified allele(s)

Reply: As commented above, please note that alleles within Blastocystis STs are underreported in the literature. So, we have focused on STs when comparing our results with those published in similar surveys.

  1. line 554: you can also refer to other bibliography, check e.g. Mohammadpour et al., 2020, Parasites and Vectors 13, Issue 1, Article number 365

Reply: The reference Mohammadpour et al. Parasit Vectors. 2020;13(1):365 has been added as per requested.

  1. Lines 558-559: Please add a reference. Also, consider that elsewhere other ST4 alleles have been found in humans (see e.g. Jimenez et al., 2019 Parasites & Vectors volume 12, Article number: 376).

Reply: Please note that current reference #58 (Stensvold et al. Infect Genet Evol 2012, 12, 263-273) is the first (to the best of our knowledge) proposing the recent entry of ST4 within human populations. Please also note that this paragraph is specifically devoted to discussing about ST4, so no other references regarding other Blastocystis STs commonly detected in humans (such as ST1-3) are needed here.